# Surface Deformation Associated with the 22 August 1902 Mw 7.7 Atushi Earthquake in the Southwestern Tian Shan, Revealed from Multiple Remote Sensing Data

**Qingyu Chen [1,2], Bihong Fu [1,*], Pilong Shi [1] and Zhao Li [1,2]**

1   Aerospace Information Research Institute, Chinese Academy of Sciences, Beijing 100094, China; chenqy@radi.ac.cn (Q.C.); shipl@radi.ac.cn (P.S.); lizhao@aircas.ac.cn (Z.L.)
2   University of Chinese Academy of Sciences, Beijing 100049, China
*   Correspondence: fubh@aircas.ac.cn

**Abstract:** The 22 August 1902 Mw 7.7 Atushi earthquake is the most disastrous seismic event in the southwestern Tian Shan. However, the spatial distribution of surface rupture zones as well as the geometric feature of surface deformation remain unclear, and the seismogenic fault is still controversial. Based on geologic and geomorphic interpretations of multiple remote sensing imaging data, high-resolution DEM data derived from UAV imaging complemented by field investigations, we mapped two sub-parallel NEE-trending surface rupture zones with a total length of 108 km. In addition, ~60 km and ~48 km surface rupture zones are distributed along the pre-existing Atushi fault (ATF) and the Keketamu fault (KTF), respectively. The surface deformations are mainly characterized as bedrock scarp, hanging wall collapse scarp, pressure ridge, and thrust-related fold scarps along the two south-dipping thrust faults, which are defined as the seismogenic structure of the 1902 Mw 7.7 Atushi earthquake. Thus, we proposed the cascading-rupture model to explain the multiple rupture zones generated by the 1902 Mw 7.7 Atushi earthquake. Moreover, the multiple advanced remote sensing mapping techniques can provide a promising approach to recover the geometric and geomorphic features of the surface deformation caused by large seismic events in the arid and semi-arid regions.

**Keywords:** surface rupture zones; 1902 Mw 7.7 Atushi earthquake; multiple remote sensing data; surface deformation; thrust fault; southwestern Tian Shan

## 1. Introduction

The intracontinental convergence zones are where compression and strain accumulate and often generate giant earthquakes [1–8]. The Atushi fold-thrust belt (AFTB) is located at the superposition area of thrusting and back-thrusting tectonics that resulted from the interaction of the Pamir, western Tarim, and southwestern Tian Shan (Figure 1a) [9–14]. It is also the front belt of active tectonics developed in front of the southwestern Tian Shan (Figure 1c) [11,15,16]. The destructive Mw 7.7 Atushi earthquake occurred in AFTB on 22 August 1902, with a focal depth of ~18 km (Figure 1c) [17], affecting an ~927,000 km² area, with more than 6000 fatalities [18,19]. However, the underdeveloped technology and inaccessible traffic condition at that age led to it being impossible to implement the post-earthquake geological investigations.

The combination of multiple remote sensing imaging data, high-resolution topographic data generated from satellite remotely sensed stereoscopic pairs, and 3D prospective images provide a powerful tool to document the active deformations and geometrical features of the surface rupture zone [20–32]. Meanwhile, it could reveal the secondary geological disasters caused by the earthquakes in the high topographic relief regions, where field geological surveys can hardly implement [7,22,33]. In recent years, the numerous studies that were based on the detailed interpretation of remote sensing images as well as

field verification, such as the 1609 Ms $7\frac{1}{4}$ Hongyapu earthquake [34], 1934 Mw 8.2 Bihar-Nepal earthquake [6], 1992 Mw 7.2 Suusamyr earthquake [1], and 2008 Mw 7.9 Wenchuan earthquake [7,25], have challenged or improved the cognition, which was proposed by traditional geological methods. These studies suggest that the rupture patterns related to the mega seismic events, which occurred in the intracontinental convergence zones, are much more complicated than what have previously been recognized [4–6,26].

In the past 40 years, numerous field studies have been carried out to reveal the seismogenic fault and surface ruptures related to the 1902 Mw 7.7 Atushi earthquake [18,35–40]. However, the seismogenic fault of the 1902 seismic event is still being debated because the complicated thrust-related fold structures developed in the AFTB, and it is difficult to trace surface ruptures in the inaccessible traffic condition [18,35,37,38,40]. Some researchers suggest that the co-seismic surface rupture zones were barely formed during the 1902 Mw 7.7 Atushi earthquake because surface deformations had been absorbed by the underground folding deformation [18,38,40]. Other researchers indicated that a ~103 km surface rupture zone is developed along the pre-existing Tashipushita-Kalatage fault, which is a thrust fault with dextral slip components [35]. Meanwhile, as there is a lack of the spatial constraints of deformation characteristics, whether the seismogenic faults of 1902 Mw 7.7 Atushi earthquake are north-dipping thrust fault [18,38], south-dipping thrust fault [35] or steep strike slip fault [37,40] are still controversial.

The spatial distribution and geometric features of the co-seismic surface deformation is a key to understanding the rupture pattern of the 1902 Mw 7.7 Atushi earthquake. Therefore, we attempt to document the spatial distribution and geometric features of co-seismic surface rupture zones based on geologic and geomorphic interpretations of the multiple remote sensing images as well as the 3D perspective images, which are generated by DEM and remote sensing data. Field investigations are also carried out to verify the results revealed from multiple remote sensing data. Particularly, the Unmanned Aerial Vehicle (UAV)-based high-resolution photogrammetry was employed to carry out the throw measurement of surface ruptures to determine the geomorphic features of seismogenic structure. Through synthetic analyses of spatial distribution of surface rupture zone and deformation features of seismogenic faults, we try to reveal the rupture pattern for the 1902 Mw 7.7 Atushi earthquake. Finally, we shall revalue the accuracy of high-resolution topographic data derived from UAV-based photogrammetry for constraining the geomorphic features of the surface rupture zone.

## 2. Tectonic Setting

The ~20 km width AFTB is located in the Kashi foreland thrust basin, which is the boundary of the tectonic deformation zone between two opposite-verging fold-thrust belts (Figure 1a,b) [11,12,14,41–47]. The NEE-trending AFTB is bounded by the north-dipping Kashi basin thrust fault (KBT) in the north, which is also called the Tuotegongbaizi-Aerpaleike fault [18,38]. Southwards, AFTB mainly consists of three-row anticlines, named Tashipisake Anticline (TPA), Keketamu anticline (KTA), and Atushi anticline (ATA), as well as three-row thrust faults with the same name as the anticlines, successively (Figure 1b,c). The north-dipping Tashipisake fault (TPF) is exposed on the south flank of the TPA. The Atushi fault (ATF) and the Keketamu fault (KTF) are dipping to the south, which are exposed on the north flank of the ATA and KTA, respectively (Figure 1b,c) [10–12,40,43,44,48–51].

Pre-Cenozoic strata are composed of Carboniferous limestone, Permian sandstone, and Cretaceous thick purplish red sandstone interbedded with grayish green siltstone, which are exposed at the hanging wall of the KBT as well as the TPF [43,48]. With the continuous southward expansion of southwestern Tian Shan since the early Miocene, the 8–10 km upward coarsening Cenozoic strata are deposited in the depression basin (Figure 1d) [11,48,52]. The Paleogene Kashi Group consists of grayish green and brownish red sandstone, siltstone, and argillite, interbedded with marine limestone and gypsum [43,53], which are locally exposed at the eastern of Tashipushika (Figure 1c).

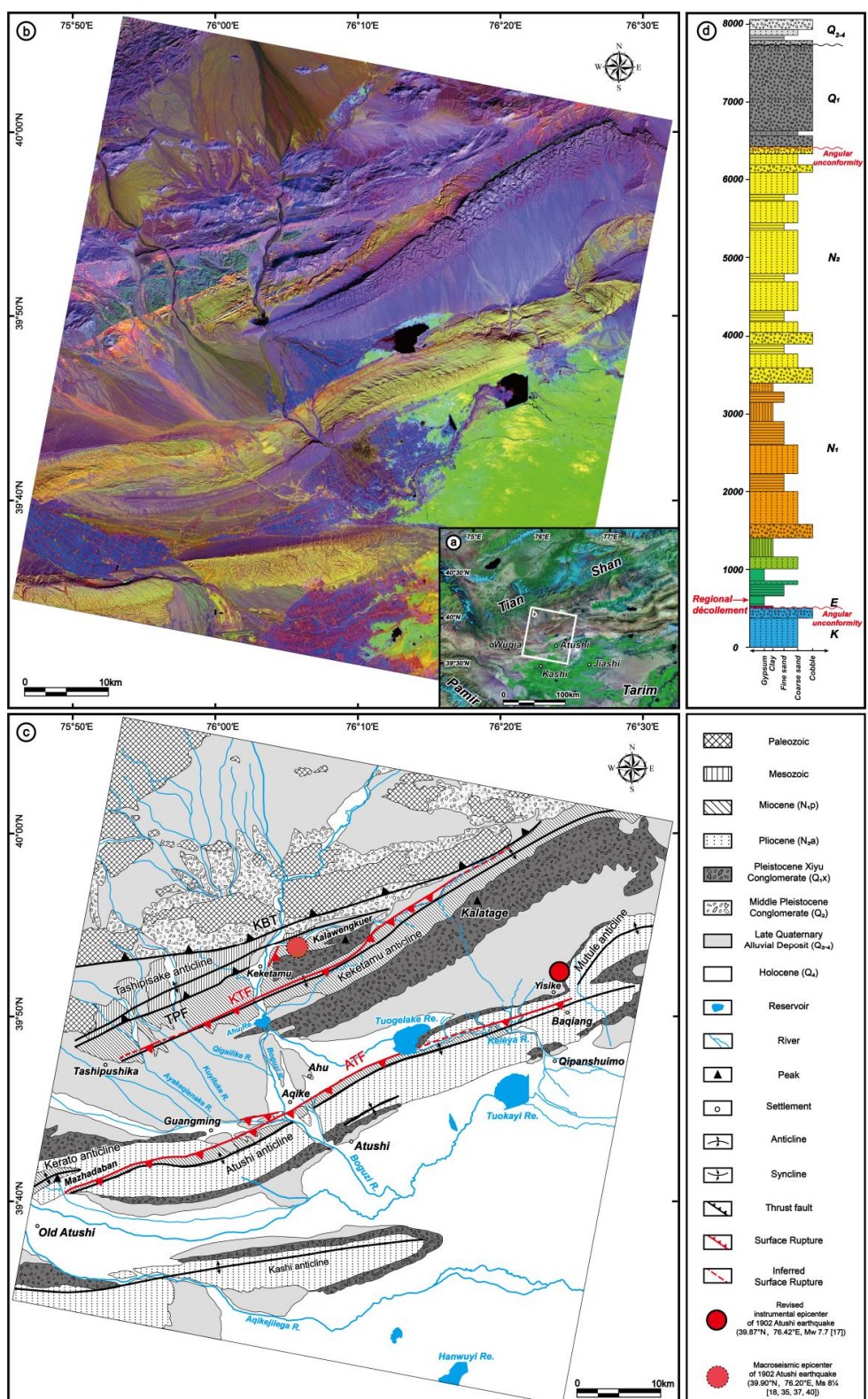

**Figure 1.** (**a**) A Landsat ETM/TM mosaic image of the Pamir-Tian Shan convergence zone, white polygon shows the location of the AFTB. (**b**) ASTER false color composite image of 2/1(R), 3(G) and 5/8(B) of AFTB; (**c**) geological interpretation of AFTB showing that there are 3–4 rows of sub-parallel active fault-related folds. Stratigraphic and tectonic data are modified from [11,12,48,50,54]. Seismic data are from [17,18,35,37,40]; (**d**) summary of AFTB stratigraphic columnar profile from Cretaceous to Quaternary, according to [11,48,53].

The Miocene Wuqia Group is mainly composed of grayish green and brownish red siltstone, argillite, interbedded with grayish yellow sandstone as well as dark gray conglomerate. Gypsum and gypsum-bearing mudstone are deposited at the bottom layer of the Wuqia Group, which are in pseudoconformity or conformity with the underlying Paleogene Kashi Group [11,48,53]. The weak layers between Paleogene Kashi Group and Miocene Wuqia Group constitute the regional detachment layers in AFTB (Figure 1d) [40,52,55–57]. The Pliocene Atushi Formation is characterized by grayish yellow sandstone interbedded with thin brownish red siltstone and argillite, containing conglomerate in the upper layer (Figure 1d) [54]. Quaternary deposits are widely developed in AFTB (Figure 1c). The early Pleistocene Xiyu Formation is composed of dark gray, gray massive conglomerate with grayish yellow sandstone, siltstone layers, or lens. The basal age of the Xiyu conglomerate in AFTB is asynchronous, which indicated the southward multi-stage propagation processes in the AFTB [12,49,54]. The semi-consolidated or unconsolidated middle Pleistocene ($Q_2$) deposits consist of the fluvial, alluvial, and lacustrine deposits. The unconsolidated late Pleistocene ($Q_3$) deposits are composed of alluvial deposits. Holocene ($Q_4$) strata include unconsolidated alluvial, aeolian, and talus deposits [11,53]. The late Quaternary sediments ($Q_{3-4}$) are mainly deposited in the streambed, terraces, and alluvial fans in AFTB (Figure 1c).

## 3. Method and Data

### 3.1. Remote Sensing Data and Image Processing Methods

In this study, we conducted geological and geomorphological interpretations of satellite imagery and topographic data, as well as field investigations. Multiple sources of remote sensing data were used including Gaofen-2 (GF2) satellite data (1-m ground resolution after fusion of multi-spectral and panchromatic image), the ASTER Visible and Near Infrared (VNIR, 15-m ground resolution) as well as short-wave infrared data (SWIR, 30-m ground resolution), Landsat TM/ETM mosaic data (30-m ground resolution), and UAV-based orthomosaic images (1–5-cm ground resolution). Multispectral satellite remote sensing images have advantages for interpreting the structural features and sedimentary units in uncovered bedrocks area [11]. For better geological interpretation and lithological extraction, the ASTER false color composite images (R: band ratio 2/1, G: band 3, and B: band ratio 5/8, Figure 1b) were used to distinguish different lithological units in this region. We also generated 3D perspective images of typical area by virtue of the ASTER multispectral images and DEM data from ALOS World 3D-30 m (AW3D30), which is beneficial for the geomorphic feature analysis of AFTB.

Surface rupture is a common transformation approach of crustal elastic strain into permanent tectonic deformation [7]. Geometry of the surface rupture zones contain reliable information for determining the seismogenic mechanism and tectono-geomorphic growth in active tectonic belts [11]. Different types of surface ruptures can be recognized along the fault depending upon the geometry of fault, as well as the geomorphic and sedimentary units where it developed. Our interpretation of co-seismic surface rupture zones associated with 1902 Mw 7.7 Atushi earthquake is based on two criteria: (1) The fault scarps break the youngest geomorphic units such as the lower terraces or alluvial fans, and (2) the faults dislocated the latest sedimentary layers.

### 3.2. Throw Measurements by the UAV

In order to measure precisely geomorphic features of co-seismic rupture zones related to the 1902 Mw 7.7 Atushi earthquake, the DJI Phantom 4 RTK UAV equipped with FC6310R camera (focal length is 8.8 mm, and sensor dimensions is 2.833 mm × 8.556 mm) was used to acquire high-resolution photography data. The flight altitude is usually lower than 120 m above the surface, so that the resolution of DSM and orthomosaic images could reach ~5 cm (Equation (1)):

$$R = H \times Sc/f \tag{1}$$

Here, *R* is the ground resolution (cm/pixel), *H* is the flight height (m), *Sc* is the pixel size (mm × mm), and *f* is the focal length (mm). DSMs, orthomosaic images, and three-dimensional tectono-geomorphic models were generated via the Pix4D software. Topographic profiles are not always precisely perpendicular to the fault scarps to ensure that profiles were derived from the same geomorphologic surface, as well as the place where erosion or degradation is minimal to reduce uncertainties in geomorphic surface shapes. In each profile derived from UAV-based high-resolution DEM data, trend lines fitted to represent for the original geomorphic surface were projected onto the fault trace to measure the vertical offset [19,27]. Other topographic profiles could be acquired from the AW3D30 in the same principle.

## 4. Results

### 4.1. Spatial Distribution and Geometric Features of Surface Rupture Zones

Based on the detailed interpretation and analyses of multiple remote sensing images and high-resolution DEM data obtained by UAV, combined with the field investigations, we found that surface rupture zones related to the 1902 Mw 7.7 Atushi earthquake distribute along the north flanks of the KTA and ATA, respectively, along the AFTB. A total length of surface rupture zones is ~108 km, which display as south-dipping, active fault and folding scarps, mainly distributing the pre-existing NEE-trending ATF and KTF (Figures 1c, 2b and 3b).

#### 4.1.1. Spatial Distribution along the KTF

From west to east, a ~48-km-long NEE-striking surface rupture zone is developed along the KTF (Figure 2). The surface rupture zone starts near Ta. village in the west, cuts through the terraces of the Ayakeqianake river, the T1 of Boguzi river to the south of Keketamu village (Figure 8e), and the alluvial terraces and fans around the Kalawengkuer. Eastward, the linear trace becomes unclear to the north of Kalatage Mountain, revealing the termination of the surface rupture zone (Figures 1c and 2a,b). In the north of Keketamu, the surface rupture zone exhibiting a NE-NEE-striking 2-km-long branch, which displaced the alluvial fans in the front of Kalawengkuer. The trace of surface rupture zone is straight in the west of Keketamu and in the east of Kalawengkuer with an NEE striking, an NE-striking bend is developing in the Pre-Quaternary sedimentary rocks, which is connecting them in the south of Kalawengkuer (Figure 2b).

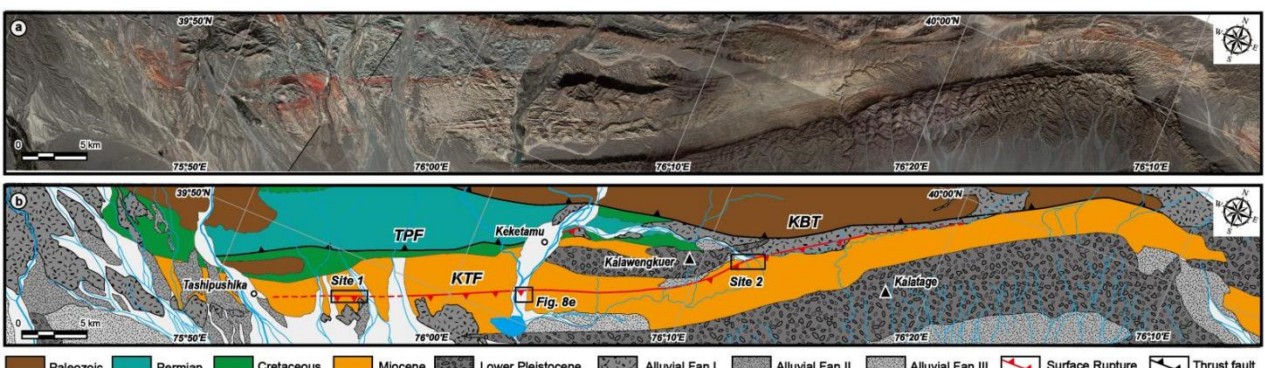

**Figure 2.** Geometric feature of surface rupture zones along the KTF. (**a**) GF-2 satellite image after fusion of multi-spectral and panchromatic images; (**b**) geological interpretation of the GF-2 satellite image showing the geometric features of the surface rupture zone developed along the KTF.

#### 4.1.2. Spatial Distribution along the ATF

Similarly, a ~60-km-long NEE-striking surface rupture zone is distributing along the ATF (Figure 3). The surface rupture zone starts near Old Atushi town in the west, cuts through the Neogene strata and the Late-Pleistocene deposits from the south of Mazhadaban to Guangming village. To the east of this village, surface rupture zones

are bifurcated into two discontinuous fault branches (Figure 3b). The southern branch of the surface rupture zone is continuously connected to the surface rupture zone in the west, but the south branch of the surface rupture zone was intensively eroded by the river, and surface rupture traces are barely preserved. The northern branch of surface rupture zone is displayed as ~3.5-km-long W-E-trending linear features that cut alluvial terraces and fans in the west of Aqike village (Figure 3a,b). Eastward, the surface rupture zone is basically developed along the boundary between the Neogene strata and Late Quaternary sediments. Meanwhile, it is also a boundary between the anticline ridge and Late Quaternary alluvial and proluvial deposits (Figures 1b,c and 3a,b). To the east of the Tuogelake reservoir, the surface rupture zone is basically developed along the streambed of the Keleya river, and the trace of surface rupture zone is not obvious because of the erosion of the river (Figure 3a,b). To the south of Yisike, the surface rupture zone disappeared in the Pre-Quaternary sedimentary rocks and terminated at the eastern end. The geometry of surface rupture zone along the ATF is displayed as a slight "S-shape" that can clearly be mapped from the satellite images (Figures 1b,c and 3a,b).

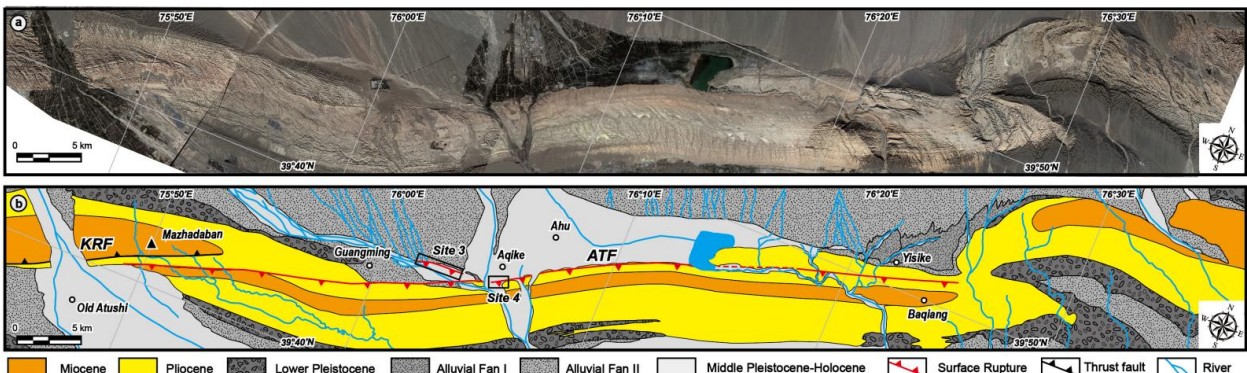

**Figure 3.** Geometric feature of surface rupture zones along the ATF. (**a**) GF-2 satellite image after fusion of multi-spectral and panchromatic images; (**b**) geological interpretation of GF-2 satellite image showing the geometric features of the surface rupture zone developed along the ATF.

### 4.2. Surface Deformation and Vertical Displacement

The western end of the KTF is located on Ayakeqianake (near Site 1 in Figure 2b). The trace of surface rupture zone in this area is relatively continuous, and it is expressed as a ~2-km-long N73°E-striking fault scarp (Figure 4a). Two alluvial terraces (labeled as T1 and T2 in Figure 4c) can be identified on the eastern bank of the Ayakeqianake river. T1 has been artificially modified, and a new fault scarp is cutting T2 (Figure 4b,d). The pale-yellow fine-grained sediments along the surface rupture may suggest a sag pond formed by the active tectonic deformation (Figure 4d). Eastward, KTF cuts the sedimentary rocks of Miocene Wuqia Group, which consists of grayish green, brownish red siltstone, and argillite.

The surface rupture zones in the east of Kalawengkuer area (as shown as Site 2 in Figure 2b) are displayed as an ~7.2-km-long fault scarp with a striking of N48°E to N60°E, which cuts $Q_{3-4}$ deposits and Miocene strata (Figure 5a,b). A distinct fault scarp is developed on the terrace surface of T2, and topographic measurements conducted by the field investigation as well as DEM data generated from the UAV photographic data show that the heights of fault scarp range from 2.2 m to 3.0 m (Figure 5c,d). The KTF is a south-dipping thrust fault with a dip angle of 28°. The fault plane in this area is developed along the detachment layer, which is comprised of grayish green, brownish red, and grayish yellow siltstone, argillite, and gypsum layers as we observed during the field investigation. A large-scale of collapse debris from the early Pleistocene conglomerate and Pliocene sandstone are distributed to the north of the Kalawengkuer Mountain. Those surface deformation and collapse debris suggest that the young fault scarps are most likely

generated by the latest large seismic event, which should be related to the 1902 Mw 7.7 Atushi earthquake.

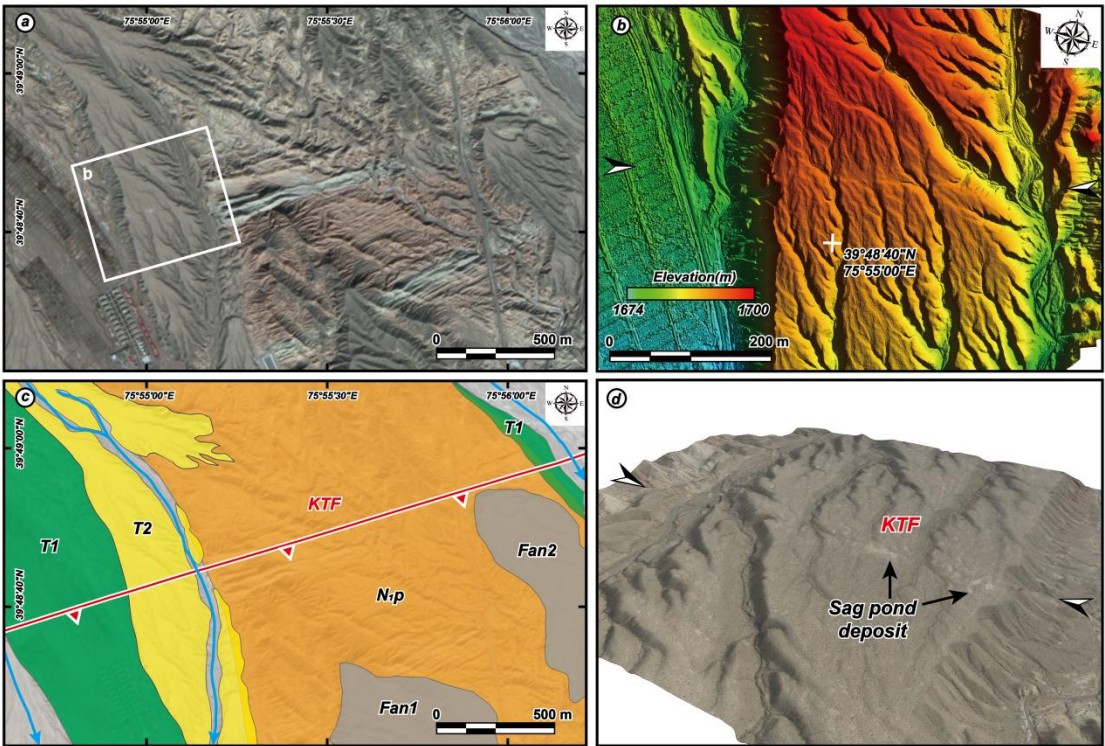

**Figure 4.** The tectono-geomorphic features in the Ayakeqianake (site 1 as shown in Figure 2b) (**a**) GF-2 satellite image showing the surface rupture break through the T2 surface of Ayakeqianake river and Miocene sedimentary rocks. (**b**) The Digital Surface Model (DSM) generated by the UAV-based photogrammetry showing a linear fault scarp was developed on the T2 surface, the black and white arrows present the trace of surface rupture; (**c**) interpretation of the GF-2 image; (**d**) The UAV Digital Orthophoto Map (DOM)-based 3D perspective image showing the geomorphic details of fault scarp along the KTF (black and white arrows), noting that the sag pound deposit in front of the fault scarp, which is most likely associated with the latest seismic event, and the view direction towards the fault scarp is shown in (**b**).

In the middle segment of ATF (as shown as Site 3 in Figure 3b), the 3.5 km-long continuous fault scarp with a striking of W-E is developed on T2 (Figure 6a,b). The topographic profiles derived from the DSM generated by UAV show that throws of fault scarps ranging from 2.2 m to 2.7 m (Figure 6c), which is consistent with the measurement of the fault scarp during the field investigation, the surface shortening in this place ranging from 4.1 m to 5.0 m. In addition, there is an eastward flowing temporary stream along the striking of fault scarp, which formed a series of erosion scarps in front of the fault scarp (Figure 6c). The difference between two kinds of scarps is the heights of scarp that formed by erosion varied along the flow direction; in contrast, the throws of the fault scarp exhibit a good consistency along the fault strike (Figure 6c).

To the south of Aqike village (site 4, Figure 3b), a 1.4 km-long fault scarp with a striking of N48°E developed on the T2 (Figure 7a,b). The surface deformation is characterized by a typical flexural structure. Although the surface of T2 has been locally modified by human activity, a linear trace of fault scarp can be clearly identified along T2 (Figure 7b). The measurements from UAV-based DSM data show that the throws of north-facing fault scarp range from 2.2 m to 2.6 m (Figure 7c), which can match with the vertical displacement measured during the field investigations, the amount of surface shortening ranging from 4.1 m to 4.9 m at this site.

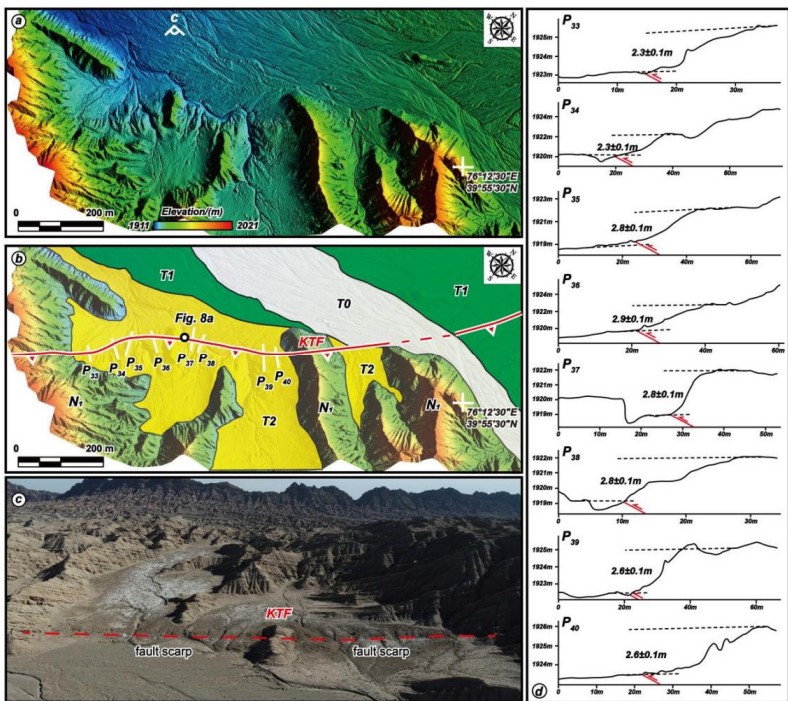

**Figure 5.** The tectono-geomorphic features in the eastern Kalawengkuer (site 2 as shown in Figure 2b)
(**a**) The DSM generated by the UAV-based photogrammetry and (**b**) geomorphic interpretation
showing that surface rupture had cut through the terraces' surface and Miocene sedimentary rocks.
(**c**) The UAV Photograph clearly presents that the linear fault scarps develop on the T2 surface, and
the red dashed line presents the trace of surface rupture; (**d**) the measured throws along the fault,
and the location of profiles is shown in (**b**).

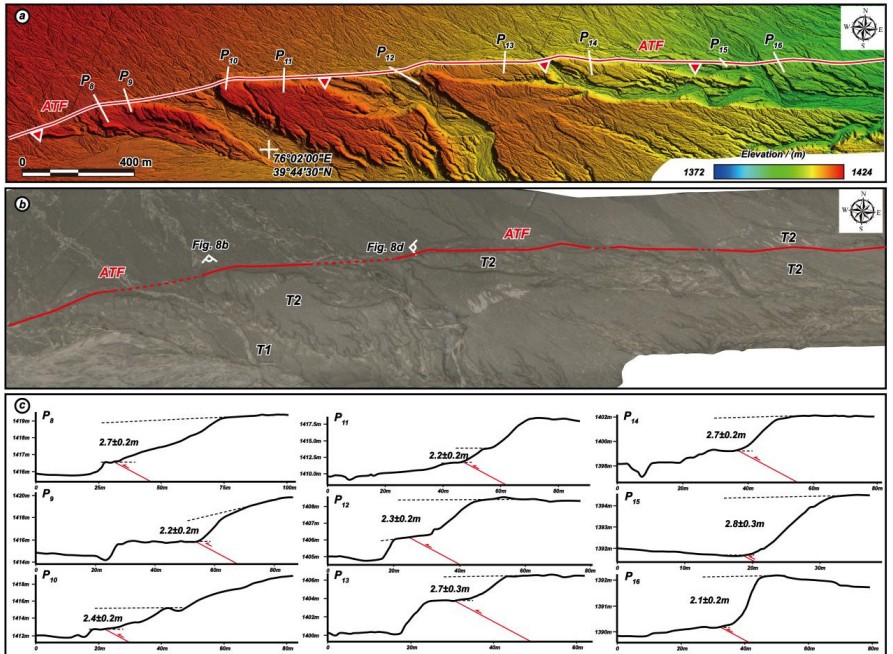

**Figure 6.** The tectono-geomorphic features in the western Aqike village (site 3 as shown in Figure 3b)
(**a**) The DSM generated by the UAV-based photogrammetry showing a linear fault scarp developed
on the T2 surface; (**b**) the UAV DOM-based 3D perspective image showing the geomorphic details
of fault scarp along the ATF (red dashed lines); (**c**) the measured throws along the fault, and the
locations of profile are shown in (**a**).

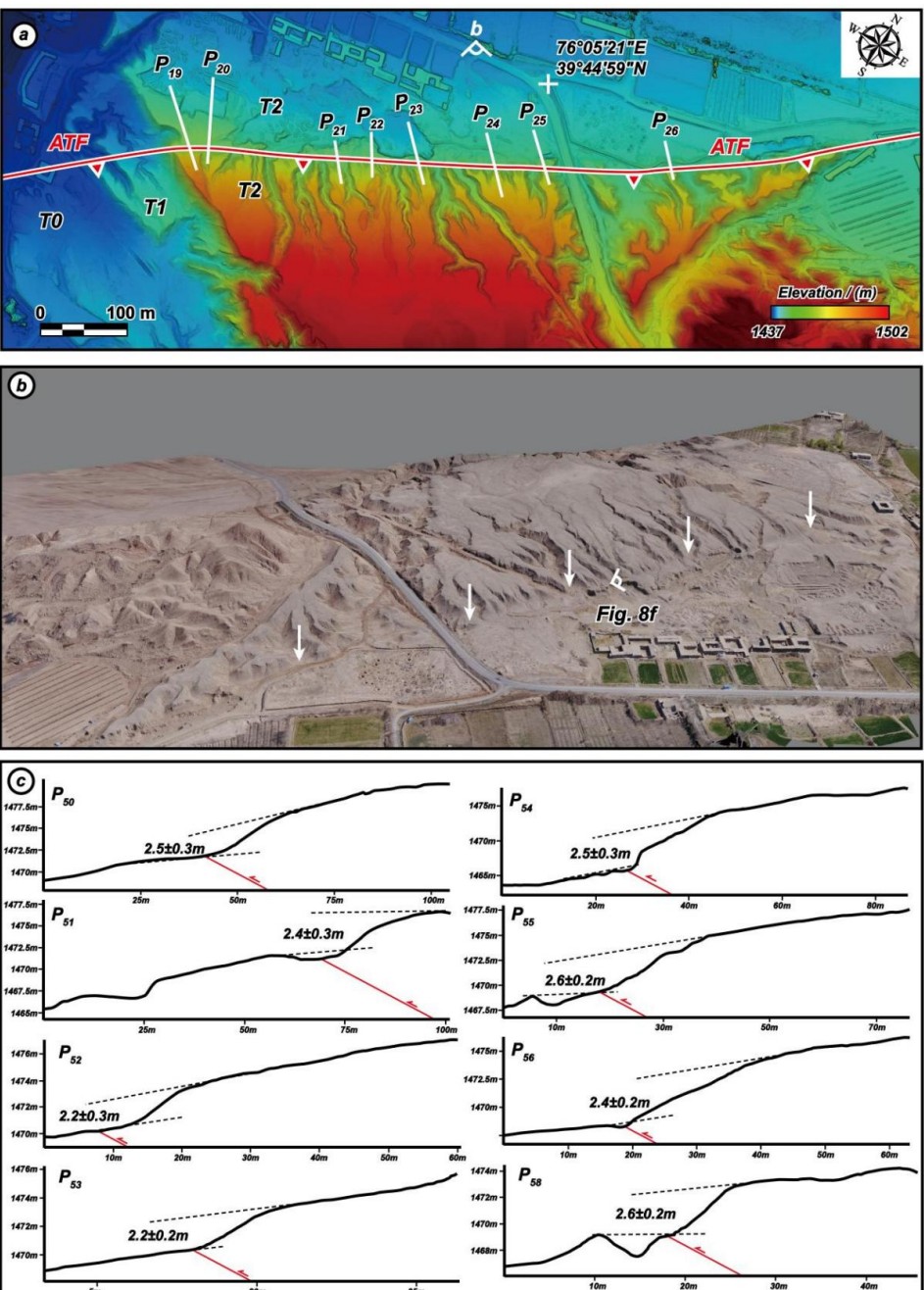

**Figure 7.** The tectono-geomorphic features in the southern Aqike village (site 4 as shown in Figure 3b) (**a**) The DSM data generated by the UAV-based photogrammetry showing linear fault scarp developed on the T2 surface; (**b**) the UAV DOM-based 3D perspective image showing the geomorphic details of fault scarp along the ATF (white arrows), the view direction towards the fault scarp shown in (**a**). (**c**) The measured throws along the fault, and the locations of the profile are shown in (**a**).

Based on throw measurement principles mentioned above, we obtained a total of 46 throw measurements of the fault scarp related to the 1902 Mw 7.7 Atushi earthquake (29 topographic profiles along the ATF, and 17 topographic profiles along the KTF), ranging from 1.4 m to 2.9 m along the ATF and KTF (Table 1). In addition to the area under intensive erosion, we try to ensure that the topographic profiles are evenly distributed along the fault. Generally, ~96% of the throws are measured with high quality and reliable credibility, while the left two measurements are rated at low credibility (marked with * in Table 1), possibly originated from the human activities. The maximum vertical displacement related

to the 1902 Mw 7.7 Atushi earthquake is 2.9 ± 0.1 m, the average vertical displacement is about 2.5 m, and the vertical displacement has good consistency on the latest alluvial deposits along the faults. The maximum co-seismic surface shortening is about 5.5 m to the east of Kalawengkuer. All of these fault scarps are featured with a hanging wall of the thrust fault in the south, indicating that the kinematic character of the fault is northward thrusting. Meanwhile, the typi-cal field photographs are displaced to show the surface rupture zones along the ATF and KTF (Figure 8).

**Table 1.** Throws measured along the ATF and KTF.

| No. | Longitude/(°) | Latitude/(°) | Distance/(km) | Throw/(m) | Error/(m) | Fault |
|---|---|---|---|---|---|---|
| 1 | 75.9553 | 39.7052 | 12.43 | 2.3 | 0.2 | |
| 2 | 75.9656 | 39.7073 | 13.38 | 2.5 | 0.2 | |
| 3 | 75.9684 | 39.7082 | 13.63 | 2.4 | 0.2 | |
| 4 | 75.9724 | 39.7096 | 14.01 | 2.3 | 0.2 | |
| 5 | 75.9768 | 39.7113 | 14.48 | 2.3 | 0.2 | |
| 6 | 75.9842 | 39.7133 | 15.12 | 2.0 | 0.2 | |
| 7 | 75.9985 | 39.7295 | 16.98 | 2.3 | 0.2 | |
| 8 | 76.0273 | 39.7383 | 20.06 | 2.7 | 0.1 | |
| 9 | 76.0283 | 39.7429 | 20.17 | 2.2 | 0.1 | |
| 10 | 76.0317 | 39.7441 | 20.56 | 2.4 | 0.1 | |
| 11 | 76.0337 | 39.7442 | 20.75 | 2.2 | 0.1 | |
| 12 | 76.0376 | 39.7447 | 21.1 | 2.3 | 0.1 | |
| 13 | 76.0413 | 39.7453 | 21.42 | 2.7 | 0.1 | |
| 14 | 76.0442 | 39.7456 | 21.68 | 2.7 | 0.1 | |
| 15 | 76.0487 | 39.7457 | 22.07 | 2.8 | 0.1 | ATF |
| 16 | 76.0507 | 39.7460 | 22.2 | 2.1 | 0.1 | |
| 17 | 76.0534 | 39.7461 | 22.5 | 1.4 * | 0.1 | |
| 18 | 76.0567 | 39.7469 | 22.8 | 1.5 * | 0.1 | |
| 19 | 76.0851 | 39.7469 | 24.6 | 2.5 | 0.1 | |
| 20 | 76.0853 | 39.7469 | 24.6 | 2.4 | 0.1 | |
| 21 | 76.0874 | 39.7475 | 24.7 | 2.2 | 0.1 | |
| 22 | 76.0875 | 39.7476 | 24.7 | 2.2 | 0.1 | |
| 23 | 76.0884 | 39.7479 | 24.8 | 2.5 | 0.1 | |
| 24 | 76.0892 | 39.7485 | 24.8 | 2.6 | 0.1 | |
| 25 | 76.0898 | 39.7489 | 24.9 | 2.4 | 0.1 | |
| 26 | 76.0908 | 39.7495 | 25.0 | 2.6 | 0.1 | |
| 27 | 76.0901 | 39.7491 | 24.9 | 2.2 | 0.1 | |
| 28 | 76.0927 | 39.7510 | 25.2 | 2.6 | 0.1 | |
| 29 | 76.4242 | 39.8462 | 55.6 | 2.5 | 0.2 | |
| 30 | 75.9580 | 39.8271 | 7.4 | 1.9 | 0.2 | |
| 31 | 76.0502 | 39.8579 | 16.0 | 2.5 | 0.2 | |
| 32 | 76.0513 | 39.8579 | 16.1 | 2.2 | 0.2 | |
| 33 | 76.1995 | 39.9204 | 33.5 | 2.3 | 0.1 | |
| 34 | 76.1998 | 39.9209 | 33.6 | 2.3 | 0.1 | |
| 35 | 76.2001 | 39.9212 | 33.6 | 2.8 | 0.1 | |
| 36 | 76.2007 | 39.9215 | 33.7 | 2.9 | 0.1 | |
| 37 | 76.2015 | 39.9220 | 33.7 | 2.8 | 0.1 | |
| 38 | 76.2016 | 39.9221 | 33.8 | 2.8 | 0.1 | KTF |
| 39 | 76.2031 | 39.9228 | 33.9 | 2.6 | 0.1 | |
| 40 | 76.2033 | 39.9229 | 34.0 | 2.6 | 0.1 | |
| 41 | 76.2255 | 39.9355 | 36.7 | 2.6 | 0.2 | |
| 42 | 76.2265 | 39.9372 | 36.9 | 2.7 | 0.2 | |
| 43 | 76.2504 | 39.9628 | 37.7 | 2.5 | 0.2 | |
| 44 | 76.2381 | 39.9421 | 38.0 | 2.7 | 0.2 | |
| 45 | 76.2579 | 39.9490 | 39.9 | 2.7 | 0.2 | |
| 46 | 76.2679 | 39.9539 | 40.9 | 2.6 | 0.2 | |

* Low credible throw measurement. Distance representing the distance from the profile location to the west end of a corresponding fault along the fault striking.

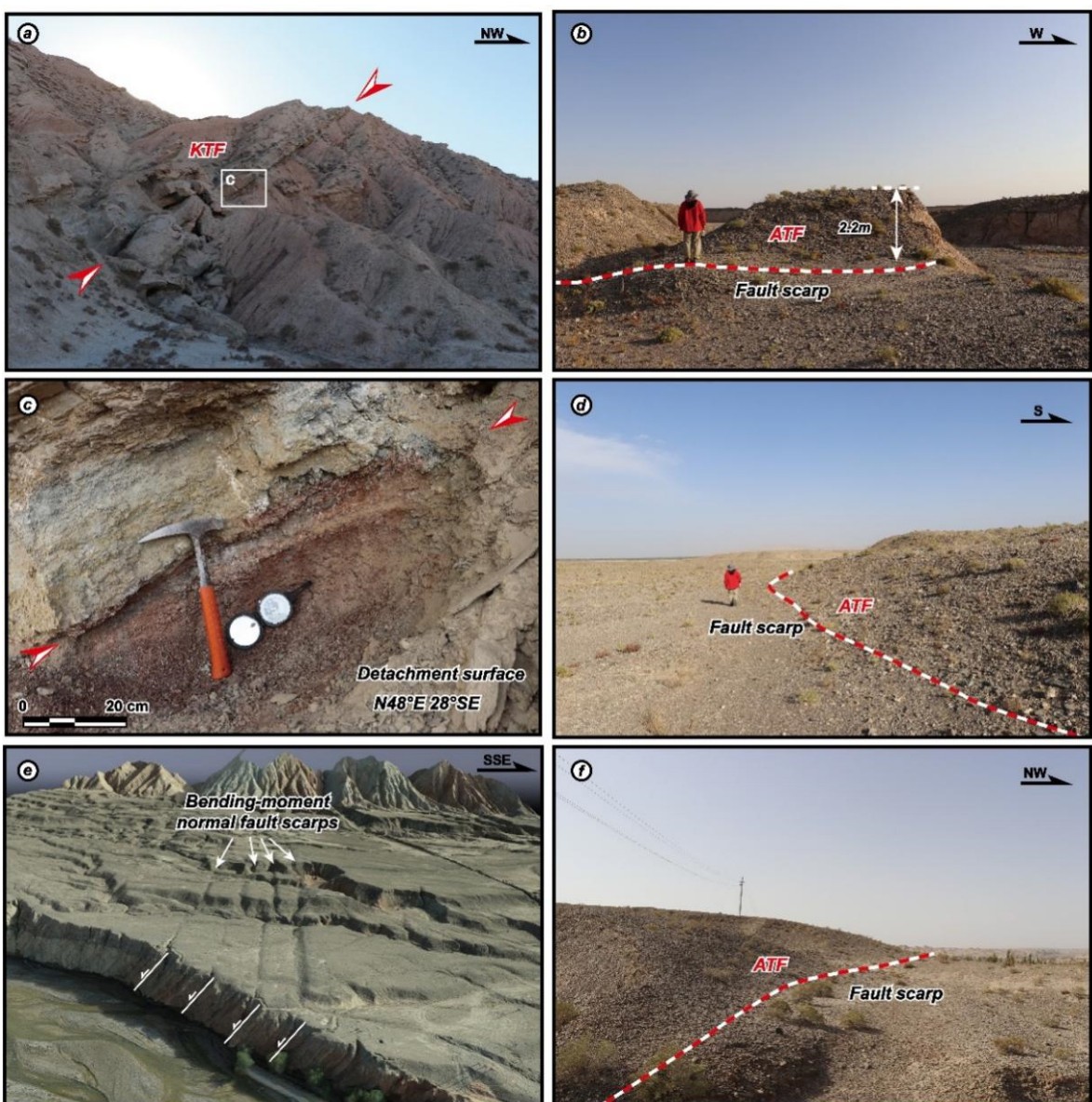

**Figure 8.** Typical field photographs showing the surface rupture zones observed along the ATF and KTF. (**a**) Photograph showing the development of active fault and collapse in Kalawengkuer, collapses are comprised of N$_1$ sandstone and siltstone. Location is shown as Site 2 in Figures 2b and 5b; (**b**) photograph showing a ~2.2-m-high fault scarp developed on T2, as a result of the 1902 Mw 7.7 Atushi earthquake. Location is shown as Site 3 in Figures 3b and 6b; (**c**) enlarged photograph showing the fault plane (as shown in (**a**)) developed in the detachment layer, which consists of brownish red and grayish yellow siltstone, argillite, and gypsum; (**d**) photograph showing the surface deformation is characterized as fault scarp developed on T2, which was produced by the 1902 Mw 7.7 Atushi earthquake. Location is shown as Site 3 in Figures 3b and 6b. (**e**) UAV-based 3D visualization of bending-moment normal fault scarps in the south of Keketamu (as shown in Figure 2b). Four north-facing scarps, which developed at the core of KTA, are sub-parallel, closely-spaced arrangement; (**f**) photograph showing the surface rupture characterized as fault scarp, which is likely related to the 1902 Mw 7.7 Atushi earthquake. Location is shown as Site 4 in Figures 3b and 7b.

## 5. Discussion

### 5.1. Geometric Feature and Surface Deformation of the Surface Rupture Zones

The co-seismic surface deformation and seismogenic fault of the 1902 Mw 7.7 Atushi earthquake is a century puzzle. There are no documents showing the surface ruptured [18,36,38,40].

Recently, Hu et al. reported that they have mapped a 103 km-long surface rupture zone along the Tashipipushita-Kalatage fault (Figure 9), based on the interpretation of satellite images combined with field investigations [35]. However, we have checked the so-called surface ruptures suggested by Hu et al. during our field investigations, and we confirmed that a part of surface ruptures indeed cut the late Quaternary deposits, but most of them are geomorphic features formed by differential weathering of Mesozoic-Cenozoic lithologic units or artificial lineaments associated with human mining activities (as shown in Figure 15 of [35]). As concerning the seismogenic fault of the 1902 Mw 7.7 Atushi earthquake, Bai and Ge (1988) and other researchers suggested that the ~200 km-long north-dipping Tuotegongbaizi-Aerpaleike thrust fault is likely the seismogenic fault (TAF in Figure 9) [18,36,38]. Meanwhile, Zhao et al. and Tian et al. inferred that the boundary fault between the southwestern Tian Shan and northern Tarim basin is the seismogenic fault, which is a left-lateral strike slip fault with high dip angle (NTF in Figure 9) [37,40,58]. However, there is no surface active deformation and geomorphic features showing the left-lateral strike slip fault. The deep seismic reflection profile also did not support any high dip angle faults developed in this region [41].

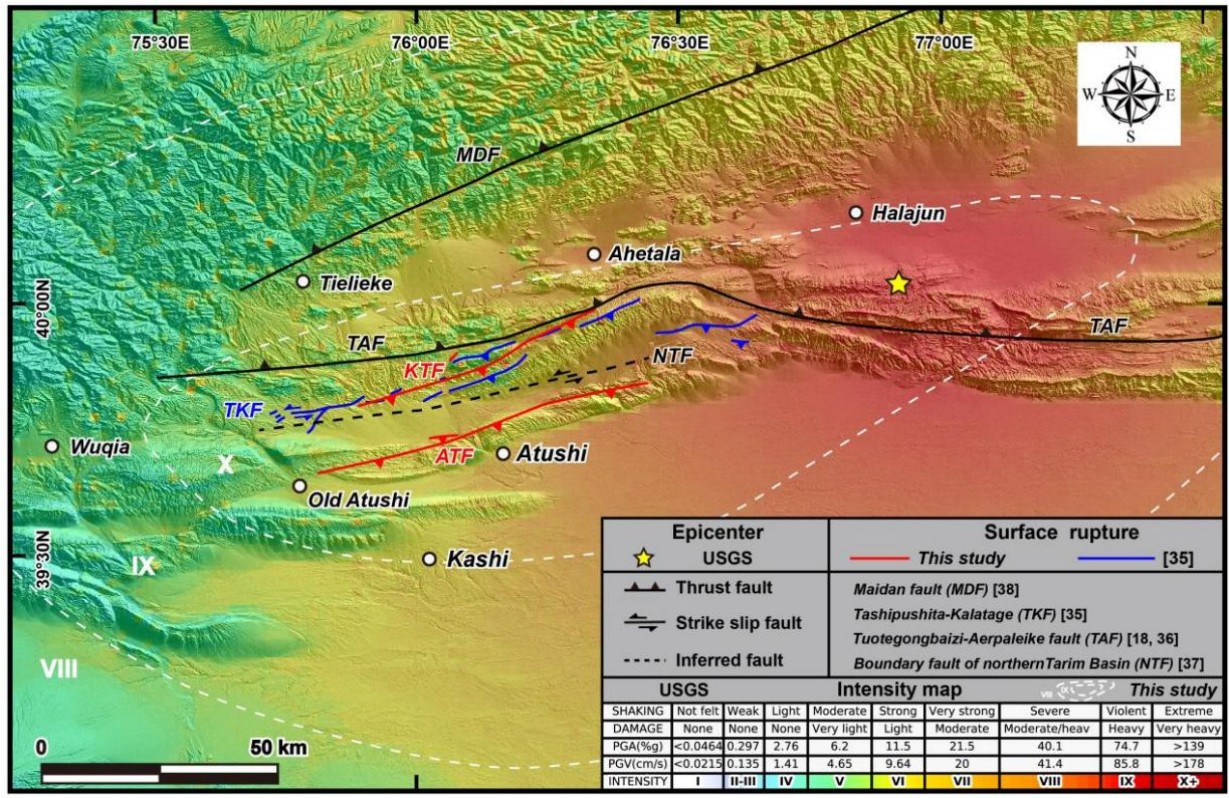

**Figure 9.** Surface rupture zones and seismogenic faults mapped by our study compared with that given by the previous studies on the 1902 Mw 7.7 Atushi earthquake. The seismic intensity map is produced based on the data from United States Geological Survey (USGS) and [18]. Red and blue lines displaying the spatial distribution of surface rupture zones revealed by this study and [35], respectively.

In this study, our new findings based on the structural and geomorphic interpretations of multiple remote sensing imaging data and high-resolution DEM data, as well as the field investigations indicated that the 1902 Mw 7.7 Atushi earthquake does break the surface, and two remarkable surface rupture zones with a total length of 108 km are clearly developed along ATF and KTF. The surface rupture zones of 1902 Mw 7.7 Atushi earthquake are mainly distributed along the pre-existing faults, which are basically developed along the northern flank of corresponding thrust fault-related anticlines. These anticlines are defended as the detachment fold, according to the previous structural analyses [43,50,54]. Tertiary argillite and gypsum layers constitute the regional decollement in the AFTB [41,52]. The geometry

of detachment layers might control the surface expression of the surface rupture zones. On the northern branch of surface rupture zones, KTF is interacted with KBT at the north of Kalatage (Figures 1c and 2b), which might control the termination of surface ruptures on the eastern end. On the southern branch of surface rupture zones, ATF is interacted with KRF around Mazhadaban (Figure 3b), which might lead the termination of the surface ruptures in the west. In the eastern part of the southern branch, the surface ruptures terminated at the east of Baqiang, which might be related to the interaction of ATF and Mutule anticline (Figure 3a,b).

The surface deformation patterns are varying along the strike of the thrust faults. Various types of co-seismic surface deformation can be identified in accordance with the geometrical features of the fault, types of surface deposits, and geomorphic units developing along the thrust faults. The high-resolution DEM data and 3D perspective generated by the UAV imaging data, as well as the field observations, revealed that there are four types of surface deformation generated by the 1902 Mw 7.7 seismic event: (1) bedrock scarp, characterized as simple scarps in Pre-Quaternary sedimentary rocks (Figure 10a); (2) hanging wall collapse scarp, where the gravitational collapse was taking place on the hanging wall of thrust faults (Figure 10b); (3) pressure ridge, due to Late Quaternary alluvial deposits flexibly deformed (Figure 10c); (4) fault-related fold scarps, featured as a series of sub-parallel, closely-spaced fold scarps generated by the fault thrusting (Figure 10d), which are quite similar with that observed at the Haermodun anticline in the front of the southern Tian Shan and Slik monocline in the front of the western Kunlun Mountain [50].

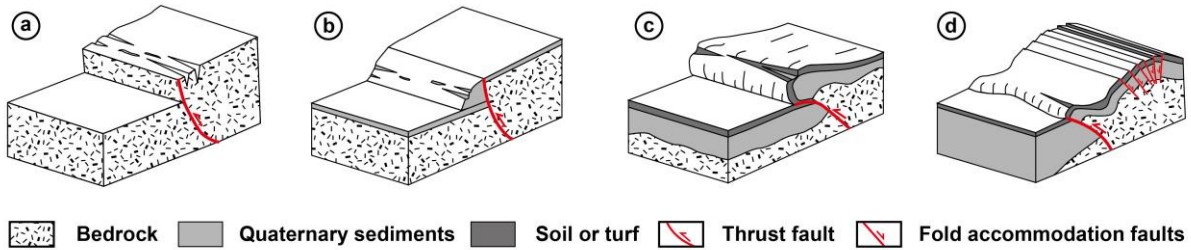

**Figure 10.** Typical surface structural deformation features developed along the ATF and KTF surface rupture zones. (**a**) bedrock scarp; (**b**) hanging wall collapse scarp; (**c**) pressure ridge; (**d**) main thrust faulting and bending-moment normal fault scarps. The moving and bending of the bed in the anticlinal hinge can amplify the bed curvature to generate bending-moment normal faults (as shown in Figure 8e) [50]. The structural models are modified after [59].

### 5.2. Rupture Length and Kinematics of the 1902 Mw 7.7 Atushi Earthquake

Rupture behavior of the intracontinental mega thrusting earthquakes was often characterized as multiple surface rupture zones like the 1950 Mw 8.7 Assam earthquake [4] and 2008 Mw 7.9 Wenchuan earthquake [7,25]. However, it is a long-unsolved problem on the surface ruptures and seismogenic fault related to the 1902 Mw 7.7 Atushi earthquake [18,35–38,40]. Mostly, it is difficult for the traditional geological investigation methods to reveal micro-geomorphic features of the co-seismic surface rupture zones over 100 years later [35,40]. However, high-resolution satellite remote sensing data, particularly the centimeter-scale UAV-based photogrammetric data, can provide us with a useful tool to map the micro-geomorphic features and spatial distribution of the surface rupture zones generated by the 1902 Mw 7.7 Atushi earthquake.

In this study, a total length of 108-km-long surface rupture zones related to the 1902 Mw 7.7 Atushi earthquake are mapped, and the kinematics are mainly characterized by thrust faulting and thrust-related folding (Figures 2b, 3b and 8a,e). The empirical equation of the relation between magnitude and rupture length of thrust faulting events (Equation (2)), which is deduced by previous studies [60,61], is as follows:

$$\log(SRL) = (0.63 \pm 0.08)M - (2.86 \pm 0.55) \tag{2}$$

Here, *SRL* is the surface rupture length, and *M* is the moment magnitude (*M* = 7.8 for the 1902 Mw 7.7 Atushi earthquake according to [17]. Based on the Equation (2) calculations, the estimated surface rupture length is ~113 km. A total of 108-km-long surface rupture zones mapped by us is approximately equal to the length of rupture zones estimated by [60,61]. Therefore, we suggest that two sub-parallel surface rupture zones distributed along the ATF and KTF are seismogenic faults of 1902 Mw 7.7 Atushi earthquake, and propose a cascade-rupturing model for this mega earthquake, which is similar to the 1950 Mw 8.7 Assam earthquake [4] and the 2008 Mw 7.9 Wenchuan earthquake [7,25].

## 6. Conclusions

The high-resolution satellite imaging techniques together with the advanced UAV-based photogrammetric approach for mapping micro-geomorphic deformation provide a powerful tool to document the spatial distribution of surface rupture zones as well as geometric and geomorphic features of surface deformations, which are generated by the large seismic events in arid and semi-arid regions like western China and Central Asia.

Our new results from geologic and geomorphic interpretations of multiple remote sensing imaging data, as well as the high-resolution DEM data and 3D perspective image derived from UAV photogrammetry, reveal that the 1902 Mw 7.7 Atushi earthquake has generated two sub-parallel NEE-striking surface rupture zones. Two remarkable surface rupture zones developed along the previous existing south-dipping thrust faults have a total length of 108 km. Surface deformations are mainly characterized by a bedrock scarp, hanging wall collapse scarp, pressure ridge, and thrust-related fold scarps developed on the Late-Quaternary alluvial deposits and Pre-Quaternary sedimentary rocks. A total of 46 topographic profiles show that the vertical displacements associated with the 1902 Mw 7.7 Atushi earthquake are ranging from 1.4 to 2.9 m (the average one is ~2.5 m) along the KTF and ATF, and the maximum co-seismic surface shortening is 5.5 m. The empirical equation calculations of surface rupture length and field investigation verified our interpreting results. Thus, we proposed a cascade-rupturing model of two south-dipping thrust faults to explain the multiple rupture zones produced by the 1902 Mw 7.7 Atushi earthquake, which is similar to the rupture pattern of other mega earthquakes such as the 1950 Mw 8.7 Assam earthquake and the 2008 Mw 7.9 Wenchuan earthquake that occurred in the intracontinental convergence zones within the Tibetan Plateau.

**Author Contributions:** Conceptualization, B.F.; methodology, Q.C. and P.S.; software, Q.C. and P.S.; validation, Q.C., P.S. and Z.L.; formal analysis, Q.C. and Z.L.; investigation, Q.C., B.F. and P.S.; resources, Q.C., B.F. and P.S.; data curation, Q.C. and P.S.; writing—original draft preparation, Q.C.; writing—review and editing, Q.C. and B.F.; visualization, Q.C. and B.F.; supervision, B.F.; project administration, B.F.; funding acquisition, B.F. All authors have read and agreed to the published version of the manuscript.

**Funding:** This work was funded by the Strategic Priority Research Program of the Chinese Academy of Sciences (XDA 20070202) and the Second Tibetan Plateau Scientific Expedition and Research Program (STEP, Grant No. 2019QZKK0901).

**Data Availability Statement:** Not applicable.

**Acknowledgments:** We gratefully acknowledge Yuan Yao of the Earthquake Agency of the Xinjiang Uygur Autonomous Region and Wen Li of the Earthquake Monitoring Center of Kizilsu Kirghiz Autonomous Prefecture for their support during field work. We are also grateful for the constructive comments and suggestions from three anonymous reviewers and academic editors.

**Conflicts of Interest:** The authors declare no conflict of interest.

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
