# Peer review of "Surface Deformation Associated with the 22 August 1902 Mw 7.7 Atushi Earthquake in the Southwestern Tian Shan, Revealed from Multiple Remote Sensing Data"

_remotesensing, doi:10.3390/rs14071663_

Round 1
Reviewer 1 Report
Chen et al analyzed the surface deformation associated with the 1902 Mw 7.7 Atushi earthquake using multiple remote sensing data and UAV images. They demonstrate the Atush earthquake ruptured the ATF and KTF for 60 and 48 km, respectively. The data are valuable and important for understanding the seismotectonics of Southern Tianshan region, which should be eventually published. The following is the main comments:
1, Since the Atushi earthquake is a thrust dominated earthquake, the DEM should be more efficient in detecting co-seismic surface deformations. The authors measured the vertical offset based on high-resolution UAV-DEM resolution ~ 5cm. However, there are only four sites covering only a small segment of the rupture zone. There are many kinds of stereo pair of remote sensing images available. Thus, I suggest the authors do derive high-resolution DEM using stereo pair of remote sensing images covering the two rupture zones along the ATF and KTF, then measure the co-seismic deformations along fault strike.
2, After obtaining the co-seismic deformation data, I suggest the authors to discuss the seismogenic structure and rupture model of the Atushi earthquake combining with previous deep structures and geological models in this region with (Zhang Xiankang et al., 2002; Tian et al., 2006).
3, The comparison with previous finds are too simple. I suggest the authors to provide the seismic intensity map and also compare with previous findings of the co-seismic surface ruptures of the Atushi earthquake. Such as, some researchers suggest the seismogenic fault is the Northern boundary fault of Tarim Basin (Tian et al., 2006). Some researchers provide detail field investigation results about the rupture (Hu et al., 2018). The authors should firstly review previoius findings about the seismogenic fault of the Atushi fault, then discuss which one is the most reliable results of the surface rupture zone and seismogenic fault. Otherwise, people maybe still trust previous Hu et al.’s results, because they conducted detail field investigations.
Overall, I suggest major revision.
Author Response
Response to Reviewer 1 Comments
Dear Reviewer,
Thank you for your constructive comments and suggestions on our manuscript entitled "Surface deformation associated with the 22 August 1902 Mw 7.7 Atushi earthquake in the southwestern Tian Shan, revealed from multiple remote sensing data". Your valuable comments and suggestions are helpful to improve the quality of our manuscript. Following your comments and suggestions we have revised manuscript carefully. The revised parts are marked in yellow for your reference. The detailed modification and change in response to your reviews are stated as following:
Point 1: Since the Atushi earthquake is a thrust dominated earthquake, the DEM should be more efficient in detecting co-seismic surface deformations. The authors measured the vertical offset based on high-resolution UAV-DEM resolution ~ 5cm. However, there are only four sites covering only a small segment of the rupture zone. There are many kinds of stereo pair of remote sensing images available. Thus, I suggest the authors do derive high-resolution DEM using stereo pair of remote sensing images covering the two rupture zones along the ATF and KTF, then measure the co-seismic deformations along fault strike.
Response 1: Deriving DEM from stereo pairs of remote sensing data is a useful tool to obtain the topographic data, especially for the regions that lacking of vegetation coverage like northwest of China (Wang et al., 2019; Liu et al., 2021). In recent years, Ren et al. (2018) reviewed the high-resolution DEM generation method from satellite remotely sensed stereoscopic pairs such as the SPOT 6/7 (~1.5 m resolution), Pleiades-1A/B (0.5 m resolution), or WorldView 1–4 (~0.5 m resolution). Thus, the resolution and precision of these DEM data are unable to measure the co-seismic deformations along fault strike because the vertical throws are only 1 to 3 meters. In the future’s study, we shall use the stereo pair of remote sensing images to derive the high-resolution DEM data with 20-30 cm resolution for mapping active deformations according to the approach given by the previous studies (Ren et al., 2016, 2018, Wang et al., 2019, 2020). Actually, based on the geologic and geomorphic interpretations of high-resolution satellite images, we have selected four typical sites to display the co-seismic surface deformation associated with the 1902 Mw7.7 Atushi earthquake, which can represent the typical and representative co-seismic surface ruptures developed on Late-Quaternary deposits along the seismogenic faults. The structural deformations recorded by the Pre-Quaternary sedimentary rocks represent the accumulative throws produced by the 1902 Atushi earthquake and previous large seismic events. It is difficult to distinguish the co-seismic deformation of 1902 Atushi earthquake from the accumulative deformations.
Point 2: After obtaining the co-seismic deformation data, I suggest the authors to discuss the seismogenic structure and rupture model of the Atushi earthquake combining with previous deep structures and geological models in this region with (Zhang Xiankang et al., 2002; Tian et al., 2006).
Response 2: According to the references recommended by you, Tian et al (2006) inferred that the seismogenic fault of 1902 Mw7.7 Atushi earthquake might be a left-lateral strike slip fault with a high dip angle based on the deep seismic reflection profile (Zhang et al., 2002). However, no surface active deformation and geomorphic features support that there is a left-lateral strike slip fault. The newly deep seismic reflection profile also did not support any high dip angle faults developed in this region (Gao et al., 2013).
In this study, our new findings based on the structural and geomorphic interpretations of multiple remote sensing imaging data and high-resolution DEM data, as well as the field investigations indicated that the 1902 Mw 7.7 Atushi earthquake does break the surface, and two significant surface rupture zones with a total length of 108 km are clearly developed along ATF and KTF. Therefore, we proposed a cascade-rupturing model of two south-dipping thrust faults to explain the multiple rupture zones produced by the 1902 Mw 7.7 Atushi earthquake, which is similar with the rupture pattern of other mega earthquakes such as the 1950 Mw 8.7 Assam earthquake and the 2008 Mw 7.9 Wenchuan earthquake that occurred in the intracontinental convergence zones within the Tibetan Plateau.
Point 3: The comparison with previous finds are too simple. I suggest the authors to provide the seismic intensity map and also compare with previous findings of the co-seismic surface ruptures of the Atushi earthquake. Such as, some researchers suggest the seismogenic fault is the Northern boundary fault of Tarim Basin (Tian et al., 2006). Some researchers provide detail field investigation results about the rupture (Hu et al., 2018). The authors should firstly review previoius findings about the seismogenic fault of the Atushi fault, then discuss which one is the most reliable results of the surface rupture zone and seismogenic fault. Otherwise, people maybe still trust previous Hu et al.’s results, because they conducted detail field investigations.
Response 3: Follow your comments, we have added a new figure showing the intensity map and seismogenic faults of 1902 Mw7.7 Atushi earthquake proposed by the previous studies (Figure 9). In this figure, we have compared the spatial distribution and geometric features of surface ruptures and seismogenic faults given by this study and previous studies.
Moreover, we have checked the so-called surface ruptures suggested by Hu et al. (2017) during our field investigations, and we confirmed that a part of surface ruptures are indeed cut the late Quaternary deposits, but most of them are geomorphic features formed by differential weathering of Mesozoic-Cenozoic lithologic units (as shown on Figures 6, 7 of their paper) or artificial lineaments associated with human mining activities (as shown on Figure 15 of their paper). Moreover, those surface ruptures are discontinuous lineaments, which are not distributed along any active faults.

Reviewer 2 Report
Manuscript focuses on Mw 7.7 Atushi earthquake co-seismic deformations observed by remote sensing data and compared with fieldwork investigations. Structure of the manuscript is correct, data presented clearly and interpretation performed with conclusions. References are cited. Material shows that remote sensing techniques as advanced satellite optical images, high-resolution DEM, UAV imaging comparing with field investigation can help to interpret surface rupture zones and calculate surface deformations of ancient earthquakes. This is very interesting material, figures are nicely presented.
My remark is: I would suggest “tectonic setting” section rather than “geological setting” in this case. And second remark is: didn’t find the depth of the quake in the text.
Basically, manuscript is ready to go.
Author Response
Dear Reviewer,
Thank you for your recognition of our manuscript entitled "Surface deformation associated with the 22 August 1902 Mw 7.7 Atushi earthquake in the southwestern Tian Shan, revealed from multiple remote sensing data". Following your comments and suggestions we have revised manuscript. The revised parts are marked in yellow for your reference. The detailed modification and change in response to your reviews are stated as following:
Point 1: I would suggest “tectonic setting” section rather than “geological setting” in this case.
Response 1: Following your suggestions, we replaced the “geological setting” by “tectonic setting”.
Point 2: And second remark is: didn’t find the depth of the quake in the text.
Response 2: Based on your comments, we added the description about the focal depth in Line 35 “with a focal depth of ~18km” according to reference [17].

Reviewer 3 Report
This manuscript provides information for mapping the fracture trail caused by the 1902 earthquake using satellite data, drone imaging, and field measurements. Comparative analysis of the data from different sources is carried out and an attempt to interpret the tectonic earth surface rupturing along several faults. The manuscript is of high interest, written transparently, and its presentation is technically correct. It is advisable to increase the size of the inscriptions in the figures for easier reading. Comments on the manuscript proposed for publication are made in the original pdf file.

Author Response
Dear Reviewer,
Thank you for your constructive comments and suggestions on our manuscript entitled "Surface deformation associated with the 22 August 1902 Mw 7.7 Atushi earthquake in the southwestern Tian Shan, revealed from multiple remote sensing data". Those valuable comments and suggestions are helpful to improve the quality of contents and figures for our manuscript. Following your comments and suggestions, we have revised manuscript carefully. The detailed corrections and change in response to your reviews are stated as following:
Point 1: It is advisable to increase the size of the inscriptions in the figures for easier reading.
Response 1: Sure, we have enlarged the size inscriptions in the figures.
Point 2: Comments on the manuscript proposed for publication are made in the original pdf file.
Response 2: Following your comments made in the pdf file, we have modified the relevant parts in the new version of our manuscript. The revised parts are marked in yellow for your reference.
- We rewrite the sentence in Line 36-38.
- In this study, we mainly used ALOS World 3D-30m data which is belong to “DEM”and DSM and DOM data are generated by UAV. So, we changed “DEM” into “topographic data” in Line 126.
- For north arrow, we deleted the north arrow in Figure 4d and 7b. We derived the DSM image using the color bar from blue to red can effectively highlight the difference in low elevation area. It is a common color bar system. Moreover, we marked the trace of surface rupture in Figure 5c.
- We described the meaning of “them”in Line 386, which is stand for [56, 57], and we also replaced “paleo-earthquake” by “large seismic events” in the new version of manuscript.
- The cascade-rupturing model is a hot research topic.
Previous studies suggested that the southward thrusting of southwestern Tian Shan induced the 1902 Mw7.7 Atushi earthquake [18, 33]. Some researchers proposed that the northward thrusting might cause the 1902 Atushi earthquake, but they did not provide any evidence on surface deformation [32, 35, 55]. In this study, our new findings based on the structural and geomorphic interpretations of multiple remote sensing imaging data and high-resolution DEM data, as well as the field investigations indicated that the 1902 Mw 7.7 Atushi earthquake does break the surface, and two significant surface rupture zones with a total length of 108 km are clearly developed along ATF and KTF. Therefore, we proposed a cascade-rupturing model of two south-dipping thrust faults to explain the multiple rupture zones produced by the 1902 Mw 7.7 Atushi earthquake, which is similar with the rupture pattern of other mega earthquakes such as the 1950 Mw 8.7 Assam earthquake and the 2008 Mw 7.9 Wenchuan earthquake that occurred in the intracontinental convergence zones within the Tibetan Plateau.

Round 2
Reviewer 1 Report
I am satisfied with the authors revison.
There is just some minor comments:In the response, the authors mentioned lots of new references which are not cited in the manuscript. These references should be also included in the main contents to make the audience to better understand the methods and the main contens of the manuscript. The readers will just read the published version of the manuscript.
Author Response
Dear Reviewer,
Thank you for your comments on our manuscript entitled "Surface deformation associated with the 22 August 1902 Mw 7.7 Atushi earthquake in the southwestern Tian Shan, revealed from multiple remote sensing data". Following your comments, we have added the description and related references to the manuscript. The revised parts are marked in red for your reference.
